# Hypoxic-Inflammatory Responses under Acute Hypoxia: In Vitro Experiments and Prospective Observational Expedition Trial

**DOI:** 10.3390/ijms21031034

**Published:** 2020-02-04

**Authors:** Tobias Kammerer, Valentina Faihs, Nikolai Hulde, Manfred Stangl, Florian Brettner, Markus Rehm, Mareike Horstmann, Julia Kröpfl, Christina Spengler, Simone Kreth, Simon Schäfer

**Affiliations:** 1Institute of Anesthesiology, Heart and Diabetes Center NRW, Ruhr-University Bochum, Georgstr. 11, 32545 Bad Oeynhausen, Germany; nhulde@hdz-nrw.de; 2Department of Anaesthesiology, University Hospital, LMU Munich, 81377 Munich, Germany; vfaihs@yahoo.de (V.F.); florian.brettner@med.uni-muenchen.de (F.B.); markus.rehm@med.uni-muenchen.de (M.R.); simone.kreth@med.uni-muenchen.de (S.K.); 3Walter Brendel Centre of Experimental Medicine, LMU Munich, 81377 Munich, Germany; 4Klinik für Allgemein-, Viszeral-, und Transplantationschirurgie, LMU Munich, 81377 Munich, Germany; manfred.stangl@med.uni-muenchen.de; 5Molecular Ophthalmology, Department of Ophthalmology, University of Duisburg-Essen, 45147 Essen, Germany; mareike.horstmann@uk-essen.de; 6Exercise Physiology Lab, Institute of Human Movement Sciences and Sport, ETH Zurich, 8057 Zurich, Switzerland; julia.kroepfl@hest.ethz.ch (J.K.); christina.spengler@hest.ethz.ch (C.S.)

**Keywords:** hypoxic-inflammatory response, hypobaric hypoxia, cytokines

## Abstract

Induction of hypoxia-inducible-factor-1α (HIF-1α) pathway and HIF-target genes allow adaptation to hypoxia and are associated with reduced incidence of acute mountain sickness (AMS). Little is known about HIF-pathways in conjunction with inflammation or exercise stimuli under acute hypobaric hypoxia in non-acclimatized individuals. We therefore tested the hypotheses that (1) both hypoxic and inflammatory stimuli induce hypoxic-inflammatory signaling pathways in vitro, (2) similar results are seen in vivo under hypobaric hypoxia, and (3) induction of HIF-dependent genes is associated with AMS in 11 volunteers. In vitro, peripheral blood mononuclear cells (PBMCs) were incubated under hypoxic (10%/5% O_2_) or inflammatory (CD3/CD28) conditions. In vivo, Interleukin 1β (IL-1β), C-X-C Chemokine receptor type 4 (CXCR-4), and C-C Chemokine receptor type 2 (CCR-2) mRNA expression, cytokines and receptors were analyzed under normoxia (520 m above sea level (a.s.l.)), hypobaric hypoxia (3883 m a.s.l.) before/after exercise, and after 24 h under hypobaric hypoxia. In vitro, isolated hypoxic (*p* = 0.004) or inflammatory (*p* = 0.006) stimuli induced IL-1β mRNA expression. CCR-2 mRNA expression increased under hypoxia (*p* = 0.005); CXCR-4 mRNA expression remained unchanged. In vivo, cytokines, receptors, and IL-1β, CCR-2 and CXCR-4 mRNA expression increased under hypobaric hypoxia after 24 h (all *p* ≤ 0.05). Of note, proinflammatory IL-1β and CXCR-4 mRNA expression changes were associated with symptoms of AMS. Thus, hypoxic-inflammatory pathways are differentially regulated, as combined hypoxic and exercise stimulus was stronger in vivo than isolated hypoxic or inflammatory stimulation in vitro.

## 1. Introduction

Mammalian responses to hypoxia in combination with either inflammation or exercise stress, are highly related and induce the same hypoxia-inducible-factor-1α (HIF-1α) signaling pathway [1,2,3,4,5,6]. In this regard, HIF-1α and its target genes are the key regulators allowing to adapt or even counteract hypoxic or inflammatory conditions [5,6].

Under normoxia, HIF-1α is constantly built and within minutes degraded by the ubiquitin–proteasome system after hydroxylation by oxygen-dependent prolylhydroxylases and recognition by the von Hippel–Lindau protein [5]. However, when oxygen partial pressure decreases, activity of HIF-degrading prolinhydroxylases (PHD) drop and HIF is stabilized, dimerizes with HIF-1β [5], and induces the transcription of HIF-1 target genes to overcome hypoxia inducing erythropoiesis, vasodilatation, vascular growth, or release of various mediators [6,7].

In this regard, previous research mainly focused on cell-cultures in vitro or experiments using animals [8,9,10,11]. Thus, it is known that a single hypoxic or inflammatory stimulus in mammalian cells in vitro induces the HIF-1α signaling pathway [8]. A preceding and repeated inflammatory stimulus can induce cellular tolerance, a phenomenon known from inflammatory disorders as lipopolysaccharid tolerance [6,7,12,13]. Tolerant cells lose their ability to induce HIF-1α mRNA expression and cellular protein content even when a strong second stimulus is given [9]. This even applies to humans in the initial phase of severe sepsis, where surprisingly, HIF-1α mRNA expression, cellular protein, and HIF-target genes are already highly decreased in leukocytes [6,7,12,14]. This inhibited HIF-pathway shows that immunosuppressive phenotypes are immediately initiated after the onset of sepsis [6,7,12,14].

Of note, there is a high interindividual variability in expression of HIF and HIF target genes following inflammatory or hypoxic stimuli, which at least partially can be explained by genetic variants like single nucleotid polymorphisms. In this context, we have previously shown that additionally to the extent of hypoxic or inflammatory stimuli or their duration, genetic variants in HIF-1, HIF-2, and HIF-degrading PHD-2 alter human adaptation to hypoxia [15,16,17,18,19]. High altitude residents thus have blunted hypoxic ventilatory response and diminished pulmonary hypertension under hypoxia [20,21], increased heart rate, improved peripheral oxygen saturation, and reduced incidence of mountain sickness, as assessed using the Lake Louise Score (LLS) [15,16,22,23,24]. LLS, which has recently been revised, is a well-established tool to evaluate the severity of acute mountain sickness (AMS) [25,26]. However, little is known about possible associations between molecular hypoxic-inflammatory responses and AMS or increased LLS, respectively.

It is known that both exercise and hypoxia can alter mRNA expression and protein release of pro- and anti-inflammatory cytokines, like interleukin-1β (IL-1β), IL-6, or IL-10, activate lymphocytes, alter chemokine receptors, or induce further signaling pathways of the hypoxic inflammatory response [27,28,29]. Especially, in high altitude cerebral edema (HACE) proinflammatory cytokines like IL-1β, IL-6, or vascular endothelial growth factor A (VEGF-A) are believed to play an important role [30,31]. Apart from pro- and anti-inflammatory cytokines, the role of chemokines is of particular importance, as chemokine expression and its ligands are crucial for immune cell migration, e.g., in neuroinflammatory processes, and could therefore also be of interest with respect to hypobaric hypoxia-induced AMS and high altitude cerebral edema (HACE) [32,33,34].

In summary, we know that the HIF-pathway is induced by hypoxia and by inflammation and exercise stimuli. However, little is known about in vivo responses to hypoxia and exercise on regulation of HIF-dependent pathways in healthy individuals, e.g., mountaineers. In this prospective observational trial, we first analyzed in vitro the effects of hypoxic and inflammatory stimuli on HIF-pathway genes and cytokines released in peripheral blood mononuclear cells (PBMCs) from healthy non-acclimatized volunteers. These volunteers then participated in an expedition trial that involved significant changes in oxygen concentrations in which we again analyzed HIF-pathway genes, inflammatory markers, clinical variables, and LLS at intervals following exposure to acute hypobaric hypoxia.

Thus, we tested the hypotheses that 1) hypoxic and inflammatory stimuli both induce hypoxic-inflammatory signaling pathways in vitro, 2) similar results are seen in vivo under hypobaric hypoxia, and 3) that induction of HIF-dependent genes is associated with AMS.

## 2. Results

### 2.1. Induction of Hypoxic-Inflammatory Pathways

In this study, blood smears were performed both at baseline (Munich 520 m above sea level (a.s.l.)) and after 24 h under hypobaric hypoxia (Little Matterhorn summit at 3883 m a.s.l.). Following hypobaric hypoxia, cells show increased amount of intracellular HIF-1α protein compared to baseline (see Figure 1). Thus, our data show, that the exposure to hypobaric hypoxia was sufficient to induce hypoxic-inflammatory pathways.

### 2.2. IL-1β

First, we analyzed the mRNA expression of IL-1β, a master cytokine of the inflammatory response, linking hypoxic and inflammatory pathways.

In vitro, compared to unstimulated controls (0.27 (0.14–0.47)), mRNA expression of IL-1β was increased both following isolated inflammatory stimulation with CD3/C28 (0.49 (0.31–0.84); *p* = 0.006) or profound hypoxia (5% O_2_: 0.39 (0.24–0.88); *p* = 0.004, Figure 2A for 24 h.

Similarly, IL-1β mRNA expression increased after 24 h at hypobaric hypoxia on the mountain (1.94 ± 0.93 vs. Munich 520 m: 1.38 ± 0.63, *p* = 0.05, Figure 2B). After 24 h at 3883 m, there was a significant association between IL-1β mRNA expression and the individuals’ LLS at this time point (*r* = 0.734, *p* = 0.03; Figure 2C).

### 2.3. Chemokine Receptors

#### 2.3.1. CXCR-4/SDF-1

In a next step, we analyzed C-X-C Chemokine receptor type 4 (CXCR-4), a lymphocyte chemoattractant receptor, and its ligand stromal cell-derived factor 1 (SDF-1), which had been reported to be altered by combined inflammatory stress and hypoxia.

In vitro CXCR-4 mRNA expression did not change following isolated inflammatory or hypoxic stimulation (all *p* = n.s.; Figure 3A). Similar results were seen for supernatant concentration of SDF-1 protein, the ligand of CXCR-4 (all *p* = n.s.).

In contrast, in vivo following combined hypoxic and exercise stress CXCR-4 mRNA expression increased (3.46 (3.05–5.98) vs. 2.23 (1.31–2.92); *p* = 0.008) and remained elevated after 24 h on the mountain (3.03 (2.08–4.16); *p* = 0.01; Figure 3B. Furthermore, CXCR-4 mRNA expression was associated with increased LLS after 24 h on the mountain (*r* = 0.796, *p* = 0.01, Figure 3C), whereas plasma concentration of the CXCR-4 ligand SDF-1 was unaltered after exercise at high altitude and even slightly declined after 24 h under hypoxia (all *p* = n.s.).

#### 2.3.2. CCR-2/MCP-1 and VEGF-A

In a next step, we analyzed the monocyte C-C Chemokine receptor type 2 (CCR-2), its ligand monocyte chemotactic protein 1 (MCP-1), and the target gene VEGF-A.

In vitro, CCR-2 mRNA expression increased following hypoxic incubation at 10% O_2_ (3.40 ± 2.60 vs. 2.67 ± 2.00, *p* = 0.04) and 5% O_2_ (4.16 ± 2.89, *p* = 0.005), but not following inflammatory stimulation with CD3/CD28 (*p* = n.s.; Figure 4A).

In contrast, MCP-1, the ligand of CCR-2, was unchanged following incubation with CD3/CD28 (*p* = n.s.) or 10% O_2_ (*p* = n.s.), and MCP-1 even decreased following profound hypoxic stimulation with 5% O_2_ (501.3 (309.7–809.4) pg/mL vs. 965.9 (549.2–1318.0) pg/mL, *p* = 0.02, Figure 4B). In a next step, we analyzed the MCP-1 target gene VEGF-A in the cell culture supernatant. With MCP-1 concentration being not induced, supernatant concentration of the MCP-1 target gene VEGF-A was unaltered as well both following hypoxic and CD3/CD28 incubation (all *p* = n.s., Figure 4C).

In vivo, both CCR-2 mRNA expression (3.28 ± 0.72 vs. 2.93 ± 0.83, *p* = 0.03, Figure 5A) and MCP-1 plasma concentration (50.10 (26.73 ± 79.64) pg/mL vs. <4.20 pg/mL, *p* < 0.001, Figure 5B) were increased after exercise under hypoxia compared to baseline. CCR-2 mRNA expression remained elevated after 24 h on the mountain (4.18 ± 0.95; *p* = 0.01; Figure 5A), whereas plasma concentration of its ligand MCP-1 already had returned to baseline after 24 h (*p* = n.s., Figure 5B). Notably, plasma concentrations of MCP-1 target gene VEGF-A were 5 to 10-fold greater than other subjects at baseline, and further increased as early as at arrival at 3883 m a.s.l, (54.74 (32.25–84.79) pg/mL vs. 22.60 (15.48–36.81) pg/mL, *p* = 0.005, Figure 5C), kept increased after exercise (42.70 (21.75–92.54) pg/mL, *p*=0.01) and returned to baseline after 24 h on the mountain.

Of note, already at baseline in two individuals VEGF-A plasma concentrations was 10-fold increased, and further increased under hypoxia. Importantly, a subanalysis excluding these volunteers did not alter the observed increase.

### 2.4. Proinflammatory Cytokines

Finally, we analyzed plasma concentrations of further pro- and anti-inflammatory cytokines.

#### 2.4.1. Interleukin-3

In vitro, IL-3 plasma concentration remained unchanged both following CD3/CD28 and hypoxic incubation (all *p* = n.s.).

In contrast, in vivo IL-3 plasma concentration (63.31 (25.45–119.10) pg/mL) increased as early as at arrival at 3883 m a.s.l. (122.10 (75.77–253.80 pg/mL, *p* < 0.001) kept elevated after exercise (152.9 (81.99–198.80) pg/mL, *p* = 0.05) and returned to baseline after 24 h on the mountain (94.38 (50.79–168.20) pg/mL; *p* = n.s.)

#### 2.4.2. Interleukin-6

In vitro IL-6 supernatant concentration increased highly significantly following stimulation with CD3/CD28 (134.10 (112.3–150.00) pg/mL, *p* = 0.002), but was unaltered following hypoxic incubation with 10% and 5% O_2_, respectively (all *p* = n.s.).

In vivo IL-6 plasma concentration was below detection threshold at all timepoints (all *p* = n.s; detection threshold 7.45 pg/mL).

#### 2.4.3. TNF-α

In vitro, TNF-α supernatant concentrations were below detection threshold following inflammatory and hypoxic incubations (all *p* = n.s; detection threshold 8.47 pg·mL^−1^).

Similarly, in vivo TNF-α plasma concentrations were below detection threshold, too (all *p* = n.s).

### 2.5. Anti-Inflammatory Cytokines

#### 2.5.1. IL-10

IL-10 supernatant concentrations from in vitro PBMC cultures increased following CD3/CD28 stimulation (13.81 ± 5.34 pg/mL vs. 7.25 ± 5.71 pg/mL), but not following hypoxic incubation (all *p* = n.s.).

In vivo, IL-10 plasma concentration (baseline <2.12 pg/mL) was greatly increased as early as at arrival at 3883 m a.s.l. (2.61 (2.12–5.69) pg/mL, *p* = 0.03), kept increasing after exercise (3.46 (2.12–6.28) pg/mL, *p* = 0.02), then returned to baseline after 24 h on the mountain (*p* = n.s.). Of interest, after 24 h the four individuals with highest LLS showed a trend towards lower IL-10 plasma concentrations (lowest LLS 4.02 (2.18–7.13) pg/mL vs. highest LLS 1.80 (0.91–2.48) pg/mL, *p* = 0.14).

#### 2.5.2. IL-1RA

In vitro, IL-1RA (Figure 6A) supernatant concentrations increased following CD3/CD28 stimulation (1499 ± 2225 pg/mL vs. 4842 ± 3063 pg/mL *p* < 0.01), but not under hypoxia (all *p* = n.s.).

In contrast, in vivo, IL-1RA plasma concentration was highly increased at arrival at 3883 m compared to baseline (556.80 ± 313.60 pg/mL vs. 211.80 ± 228.40 pg/mL, *p* = 0.001), kept increased after exercise (644.30 ± 428.20 pg/mL, *p* = 0.004), and after 24 h on the mountain (455.10 ± 274,40 pg/mL, *p* = 0.002; Figure 6B). After 24 h at 3883 m, the four individuals with lowest LLS showed doubled IL1-RA plasma concentrations compared to those with highest LLS, however the difference did not reach statistical significance (lowest LLS 630.5 ± 177.8 pg/mL vs. highest LLS 332.3 ± 101.7 pg/mL, *p* = 0.20).

## 3. Discussion

In this prospective observational trial, we have shown that the hypoxic-inflammatory response is differentially regulated following hypoxic or inflammatory stimulation in vitro compared to the in vivo response during an expedition in acute hypobaric hypoxia. Though, the combined hypoxic and exercise stimulus in vivo had a different and more pronounced effect on gene regulation and cytokine release than isolated hypoxic or inflammatory stimulation of PBMCs from these volunteers in vitro. Of note, both pro-inflammatory IL-1β and chemoattractant receptor CXCR-4 mRNA expression changes were associated with increased LLS and symptoms of AMS in vivo.

First, we performed in vitro experiments with PBMCs from healthy non-acclimatized volunteers using isolated inflammatory (CD3/CD28) or hypoxic (10% O_2/_5% O_2_) stimuli. Here, an isolated inflammatory stimulus increased the mRNA expression of IL-1β, supernatant protein concentrations of its antagonist IL-1RA, of pro-inflammatory IL-6 and anti-inflammatory IL-10. This is of interest, as even a slight increase in IL-1β is sufficient to cause fever, activate neutrophils, or to induce a proinflammatory response in terms of i.e., IL-6 release [35]. More importantly, recent publications suggest that in addition to pro-inflammatory mechanisms, IL-1β is responsible for the resolution of inflammation and the reprogramming of monocytes towards an immunosuppressive phenotype [36]. In this regard, our results show that CD3/CD28 induced IL-1β mRNA expression and the release of both pro- and anti-inflammatory cytokines in vitro, whereas hypoxia did not increase supernatant concentrations of aforementioned cytokines.

In vivo, we obtained similar results when we analyzed blood samples withdrawn from the same individuals after 24 h under hypobaric hypoxia. Here, IL-1β mRNA expression was increased and this was associated with an increased LLS. Thus, one might emphasize that alteration in serum cytokine concentrations under hypoxia might be associated with clinical symptoms of AMS: this is in accordance with previous studies showing that IL-1β, IL-6, and TNF-α were increased in AMS patients, or that an inhibition of anti-inflammatory IL-10 was associated with AMS in a genome wide study [31,37,38]. In contrast, other studies did not find an association between circulating cytokines like IL-3, IL-6, or IL-10, exercise and AMS [38,39,40].

In conclusion, contradictory studies had been published to this topic, which suggests that further pathways are involved in the humans’ response to acute hypobaric hypoxia and the development of AMS.

In this regard, in a next step, we analyzed chemoattractant receptor CXCR-4, and its ligand SDF-1. CXCR-4 plays an important role in target-oriented migration of stem and immune cells towards the site of infection, while recent studies even propose an association with neuroinflammation [32,33,41]. This is of particular interest as CXCR-4 and its ligand SDF-1 are the key elements regulating the entrance of lymphocytes to the central nervous system [42]. Now, we have shown an association between CXCR-4 mRNA expression and increased LLS after 24 h at hypobaric hypoxia. Further studies are needed to analyze whether the CXCR-4/SDF-1 signaling pathway is induced in AMS and HACE and whether this is associated with increased passover of lymphocytes through the blood–brain barrier [34].

In addition to CXCR-4 and its ligand SDF-1, another monocyte chemoattractant protein CCR-2 and its ligand MCP-1 are involved in neuroinflammatory diseases as well. This is highly interesting as it is known that the CCR-2/MCP-1 axis is activated by hypoxia through the HIF-1 signaling pathway [43]. Here, we were able to show that CCR-2 is induced by hypoxia but not inflammatory stimulation in vitro and increases in volunteers exposed to hypobaric hypoxia in vivo. Furthermore, its ligand MCP-1 was highly induced in our volunteers as early as at arrival in high altitude. This is interesting as the CCR-2/MCP-1 signaling pathway is induced in neurological disorders like epilepsia, and associated with neuroinflammation [44]. Furthermore, CCR-2 stimulation activates STAT3 (signal transducer and activator of transcription 3) phosphorylation and further increases IL-1β release itself, which might perpetuate inflammatory pathways aggravating AMS under persistent hypoxia [44]. Supernatant concentration of the HIF-target gene VEGF-A was unaltered in vitro after 24 h of 10% and 5% O_2_ hypoxic incubation. This is in accordance with a recent publication showing that VEGF-A increased as late as after 48 h when extreme (1% O_2_) hypoxia was applied [45,46]. In contrast, in humans, in vivo moderate hypoxia, especially when combined with exercise, is capable of increasing VEGF-A protein [40,47].

Regulation of hypoxia-induced pathways is highly complex, thus additionally to the targets analyzed in our trial, different proteins are of particular importance. In this regard, especially vasodilatator nitric oxide is of special interest, as its effects are enhanced under hypoxia as has already been shown in animal and human studies [48].

HIF-1α itself plays an important role in the modulation of CD4^+^ T cell functions under hypoxic conditions in colon cancer mice cell cultures and can decrease T cell immunity [49]. This HIF-1α-related immunosuppression can inhibit anti-tumor effects and give strong evidence for supplemental oxygen application in cancer therapies [50]. The hypoxia-HIF-related inhibition of innate immune cells may also affect the inflammatory response to hypobaric and normobaric hypoxia in our trial.

## 4. Materials and Methods

After approval by the Ethics Committee of the University of Munich, Germany (19th August 2016; project no. 350-16), healthy volunteers were enrolled in this prospective observational trial. The trial consisted of two parts: First, in vitro, we analyzed the effect of hypoxic and inflammatory stimuli on peripheral blood mononuclear cells (PBMCs) from healthy volunteers. Second, we performed a mountain expedition in which the aforementioned volunteers were acutely exposed to hypobaric hypoxia.

For in vitro experiments, whole blood samples were taken at our molecular laboratory in Munich, Germany (520 m a.s.l.) and immediately processed as depicted below. Subsequently, healthy volunteers took part in an expedition from Munich to Zermatt and blood was withdrawn and stored at three different time points as depicted in Figure 1. Detailed study protocol and results of hemodynamic changes, cerebral oxygenation, and cognitive function were published previously [24].

### 4.1. Volunteers Characteristics

Following written informed consent, healthy female (*n* = 5) and male (*n* = 6) individuals were included in this study. All subjects were in good physical and mental condition, trained, and without any comorbidities or medication. Volunteer characteristics are depicted in Table 1. Volunteers aged 18 years or older were eligible for study inclusion. Exclusion criteria included mental disorders, infection, immunological disorders, pregnancy, any kind of preexisting cardiopulmonary disease, and exposition to altitudes higher than 2000 m a.s.l. within two months before the study. During the study, one volunteer met exclusion criteria and thus was excluded from further analyses. After 24 h in hypobaric hypoxia, 6 of 11 subjects (54.6%) developed AMS (moderate or severe headache in combination with LLS point sum ≥3). Detailed volunteer characteristics are displayed in Table 1.

### 4.2. In Vitro Experiments

#### 4.2.1. PBMC Isolation

For in vitro experiments, heparinized venous blood samples from 10 of the 11 volunteers were collected and PBMCs were separated using Leucosep^TM^-Tubes (Greiner Bio-One GmbH, Frickenhausen, Germany) and Histopaque^®^ density gradient centrifugation (Sigma Aldrich, Taufkirchen, Germany) [51]. In 6-well plates, 3.0 × 10^6^ cells per well were suspended in 2 mL RPMI-medium with 10% fetal calf serum, 1% L-Glutamin (Biochrom AG, Berlin, Germany), and 1% HEPES (Sigma Aldrich, Taufkirchen, Germany) for 24 h at 37 °C with 5% CO_2_.

#### 4.2.2. Cell CULTURE Experiments

Cells were cultured at four different conditions for 24 h: a) “normoxic control”: normoxia with 21% O_2_, 5% CO_2_, 37 °C; b) “CD3/CD28”: 21% O_2_, 5% CO_2_, 37 °C with 1 mL of supernatant from PBMCs of the same subject previously stimulated with Dynabeads^TM^ Human T-Activator CD3/CD28 (Gibco, Thermo Fisher Scientific; Vilnius, Lithuania) (see Appendix A); c) “10% O_2_”: hypoxia with 10% O_2_, 40 mmHg CO_2_, 37 °C or d) “5% O_2_”: hypoxia with 5% O_2_, 40 mmHg CO_2_, 37 °C, carried out in modular incubator chambers (Billups-Rothenberg Inc, Del Mar, USA). Due to a broken seal in the hypoxic chamber of volunteer 3, PBMC analyses under 10% O_2_ could only be carried out in 9 volunteers. Hypoxia with 10% O_2_ was chosen to expose PBMCs to a similar oxygen level as at 3883 m a.s.l. in the in vivo trial. This was calculated using the alveolar gas equation, originally described by Fenn et al., and the international barometric altitude formula.

Alveolar gas equation [52]:pAO_2_ = (p_atm_ − p_H20_) × FiO_2_ − (p_A_CO_2_/RQ)
International barometric altitude formula [53]:p_atm_ = 1013.25 hPa × [1 − (6.5 × altitude/288150 m) ]^5.255^
pAO_2_ = alveolar partial O_2_ pressure; p_atm_ = atmospheric pressure; p_H20_ = water vapor pressure at 37% and full saturation p_H20_ = 47 mmHg; FiO_2_ = inspiratory O_2_-concentration; p_A_CO_2_ = alveolar partial CO_2_-pressure (normal p_A_CO_2_ = 40 mmHg); RQ = respiratory quotient (at normal nutrition RQ = 0.8).

Calculation of oxygen level for PBMCs:p_atm_ = 625.9 hPa = 469 mmHgSea level: 1013 hPa, FiO_2_ = 0.21Little Matterhorn summit (3883 m a.s.l.): FiO_2_ = 625.9 hPa/1013 hPa × 0.21= 0.13.
Alveolar gas equation:
pAO_2_ = (469 mmHg–47 mmHg) × 0.13 − (40/0.8) = 4.86 mmHg
Proportion of p_atm_: 4.86 mmHg/469 mmHg = 0.01036 = 10.36%.

Total mRNA was isolated using mirVana^TM^ miRNA Isolation Kit (Invitrogen, Thermo Fisher Scientific Baltics UAB; Vilnius, Lithuania) and TURBO DNA-free^TM^ Kit (Ambion^®^, Life Technologies^TM^, Carlsbad, USA). NanoDrop 1000 spectrophotometer (Peqlab Biotechnologie GmbH, Erlangen, Germany) was used to measure RNA quantity and purity. mRNA was stored by −80 °C till further processing and analysis. Furthermore, supernatant was stored by −80 °C for further analyses.

### 4.3. Expedition Trial

During the expedition trial, blood withdrawal was performed at four different time points (Figure 7). In detail, after initial baseline measurement in the morning in Munich at 520 m a.s.l. (Munich 520 m) all individuals were transferred to Zermatt, Switzerland, by car. In the next morning, ascent to the Little Matterhorn summit at 3883 m a.s.l. was done by funicular (duration 45 min), followed by further measurements immediately after ascent to 3883 m (24 h after baseline measurement). After this, all subjects performed endurance exercise by descending and re-ascending a ski slope between 3883 and 3500 m a.s.l. over 120 min [24]. Immediately after physical exercise, measurements were performed in an expedition tent (Keron 4 GT, Hilleberg AB, Frösön, Sweden) on the glacier (Exercise at 3883 m) at minus 11 °C. After spending one night at 3883 m a.s.l. and 24 h of hypoxic exposure, blood was taken a last time (3883 m after 24 h). Except for the third time point (Exercise at 3883 m), measurements were carried out in a protected environment (University Hospital Munich and Lodge Matterhorn Glacier Paradise). Nine mL blood was collected in EDTA-plasma tubes (Sarstedt, Nuernbrecht, Germany) at all timepoints, centrifuged at 3000 rpm for 10 min and plasma was immediately stored at −20 °C. After arrival in Munich, samples were transferred to a −80 °C freezer. Furthermore, for mRNA analyses 9 mL blood was withdrawn in PAXgene^®^ Blood RNA Tubes (PreAnalytiX^®^, Hombrechtikon, Switzerland), handled following the manufacturers protocol and stored at −20 °C. mRNA was extracted using PAXgene^®^ Blood miRNA Kit (PreAnalytiX^®^, Hombrechtikon, Switzerland) according to the manufacturer’s instructions and as described above, RNA quantity and purity was measured with a NanoDrop-1000 spectrophotometer (Peqlab Biotechnologie, Erlangen, Germany). If the amount of RNA processed was less than 8 ng/µL, an analysis could not be carried out. mRNA was stored at −80 °C until further processing.

### 4.4. Molecular Analyses

#### 4.4.1. HIF-Visualization

Isolation of HIF protein from whole blood samples is almost impossible due to short halftime of HIF proteins. Thus, we recently established the visualization of HIF protein using blood smears as published previously [6]. In detail, blood smears were prepared, and immediately frozen within 5 min after blood withdrawal. For immunofluorescence staining, blood smears were thawed for 10 min at room temperature and fixed with ice cold methanol/acetone (1:1) for 10 min at −20 °C. Afterward, immunofluorescence staining was performed using a primary mouse antihuman HIF-1α antibody (Transduction Laboratories), followed by an IgG Alexa-Flour 488 coupled goat anti-mouse antibody (Molecular Probes, order No. A11001, Eugene, OR, USA), as described previously [6].

Fluorescence microscopy was performed on an Olympus BX51 Upright (40× Lens (400× magnification); manual exposure 348 ms; FITC filter; program: cellSens Dimension) using the NIS-Elements F.30.0 imaging software (Laboratory Imaging, Prague, Czech Republic). All slides were analyzed using a standardized procedure. Erythrocytes, which account for the major portion of cells in blood smears, were easily identified due to their homogenous size (approximately 8 μm), central concavity, and typical non-fluorescent background staining pattern. All other cells with a size of 8–12 μm were analyzed, and categorized as HIF-1α positive (entirely fluorescent cells), intermediate (partly fluorescent cells), or negative (nonfluorescent cells) using the image software Image J 1.43r (National Institute of Health, Bethesda, MD, USA).

#### 4.4.2. Real-Time PCR (qPCR)

cDNA was synthesized using SuperScript^®^ III Reverse Transcriptase (InvitrogenTM, Carlsbad, USA), oligo-dT and random hexamer primers (Qiagen GmbH, Hilden, Germany), and Deoxynucleoside Triphosphate Set (dNTPs) (Roche Diagnostics, Mannheim, Germany).

Real-time PCR (qPCR) was performed in duplicates on a LightCycler 480 (Roche Diagnostics, Mannheim, Germany) with FastStart Essential DNA Probes Master, Real Time Ready Single Assays (Roche Diagnostics, Mannheim, Germany), oligonucleotides (Metabion International, Planegg, Germany), and UPL-Probes (Roche Diagnostics, Mannheim, Germany). mRNA analyses were performed at baseline (Munich 520 m); after exercise (Exercise 3883 m) and after 24 h under hypoxia (3883 m after 24 h). Primer sequences and characteristics of probes and single assays are given in Table 2. The calculation of relative mRNA expression against the reference genes TBP (TATA-box binding protein) and SDHA (Succinate dehydrogenase complex, subunit A) was performed using Advanced Relative Quantification with the LightCycler 480 Software (Roche Diagnostics, Mannheim, Germany) as described previously [54].

#### 4.4.3. Cytokine Measurements from Plasma and Supernatant

Cytokine concentrations were assessed from EDTA-plasma and cell culture supernatant using the Procarta multiplex cytokine kit (Affymetrix, Santa Clara, CA, USA) following the manufacturer’s protocol as published previously [6,23]. In detail, plasma and supernatant concentrations of proinflammatory cytokines (Interleukin-3 (IL-3), Interleukin-6 (IL-6), tumor necrosis factor α (TNFα), anti-inflammatory cytokines (Interleukin-1 receptor antagonist (IL-1RA), Interleukin-10 (IL-10)), chemokines (chemokine C-C motif ligand 2/monocyte chemotactic protein 1 (CCL-2/MCP-1), C-X-C motif chemokine 12/stromal cell-derived factor 1 (CXCL-12/SDF-1α), and vascular endothelial growth factor A (VEGF-A) were measured at all four timepoints [55]. Duplicate determinations were carried out for all measurements. The measurements were evaluated with the xPONENT^®^ software for Luminex^®^ 200 TM Version 3.1.

### 4.5. Clinical Analyses

#### Acute Mountain Sickness

Symptoms of acute mountain sickness (AMS), consisting of headache, gastrointestinal problems, insomnia, fatigue, and dizziness, were evaluated using a self-reported questionnaire according to the 2016 Lake Louise Score (LLS, 5 items, maximum point sum 15) [12]. The presence of AMS was defined as moderate or severe headache combined with LLS point sum of ≥3.

### 4.6. Statistical Analysis

Shapiro–Wilk test was performed to test for normal distribution of all datasets. Thus, normally distributed data are given as mean and standard deviation whereas non-normally distributed data are given as median with interquartile range. Changes in LLS were analyzed with Wilcoxon matched-pairs signed rank test. For molecular results, paired Student’s *t*-test was performed in case of normal distribution, whereas for non-parametric data, Wilcoxon matched-pairs signed rank test was used. Differences between groups were calculated with unpaired *t*-test or Mann–Whitney *U* test. Correlations were calculated using Spearman´s rank correlation coefficients. Every volunteer was marked with individual colors in all figures to facilitate identification. A *p*-value of ≤0.05 was considered statistically significant, and all analyzes were performed using PRISM version 7 (GraphPad Software Inc., La Jolla, CA, USA).

## 5. Limitations

Our study has limitations: First, our collective of 11 healthy female and male volunteers can be considered small and the influence of sex hormones has not been studied. However, volunteers were carefully selected, had no exposure to altitudes higher than 2000 m a.s.l. within two months prior to experiments, and both in vitro and in vivo analyses have been performed using blood samples from the same individuals. Second, in vitro, we analyzed inflammatory and hypoxic stimuli separately. Whereas, during the in vivo expedition trial on the mountain, we analyzed mRNA expression and serum cytokine concentrations from all individuals at different time points, but the study protocol did not allow to differentiate between hypoxia and exercise stress, as all volunteers underwent endurance training on the mountain. Third, on the mountain, we analyzed mRNA expression from whole blood samples and measured cytokine serum concentrations at different time points but did not isolate PBMCs in vitro to analyze ex vivo stimulability of cells exposed to hypoxia in vivo. This would be of interest to compare stimulability of PBMCs isolated in Munich under baseline conditions to hypoxic cells isolated on the mountain. However, it is known that neutrophils, which are missing in PBMCs, can be mobilized from the whole blood in case of inflammation, which is also known as leukocyte adhesion cascade [56,57]. These neutrophils are able to upregulate CXCR4 expression, which was also measured in our trial [58]. This could be a possible explanation, why CXCR4 mRNA expression was unchanged in PBMCs in vitro, whereas it was upregulated in whole blood samples in vivo. Furthermore, neutrophils are efficiently recruited under hypoxic conditions and promote angiogenesis, which could interfere with our VEGF-measurements [59]. Unfortunately, isolation of PBMCs was not possible due to technical equipment available in the glacier hut. Fourth, we performed the in vitro experiments under other climatic conditions on the mountain than in the experimental laboratory. However, the chosen oxygen level of 10% O_2_ was calculated using established formula to simulate comparable hypoxic conditions as in our in vivo trial (see “Materials and Methods”). The oxygen level of 5% O_2_ was chosen to further enhance the hypoxic effects in vitro, as it is known that PBMCs are exposed to more pronounced hypoxia within tissues or lymphatic tissue in vivo. Finally, as the study was performed in 2016, we used the unrevised LLS, which was valid at time of the study, which included insomnia. Thus, it remains unclear whether similar results can be obtained when the 2018 LLS is used [25].

## 6. Conclusions

In this prospective observational in vitro and in vivo trial, we have shown for the first time that the hypoxic-inflammatory response is differentially regulated following hypoxic and inflammatory stimulation in vitro and in vivo. In detail, exposure of healthy volunteers to acute hypobaric hypoxia with endurance exercise had a different and more pronounced effect on mRNA expression and cytokine release than isolated inflammatory or hypoxic incubation of PBMCs in vitro. Of note, both pro-inflammatory IL-1β and chemoattractant receptor CXCR-4 mRNA expression were associated with increased LLS and symptoms of AMS. Further studies are needed to elucidate the pathophysiological pathways linking IL-1β, CXCR-4, and CCR-2 signaling pathways and risk for AMS. Furthermore, a study is planned to analyze whether PBMC stimulability differs between cells isolated under normoxia and cells isolated following 24 h under hypobaric hypoxia.

## Figures and Tables

**Figure 1 ijms-21-01034-f001:**
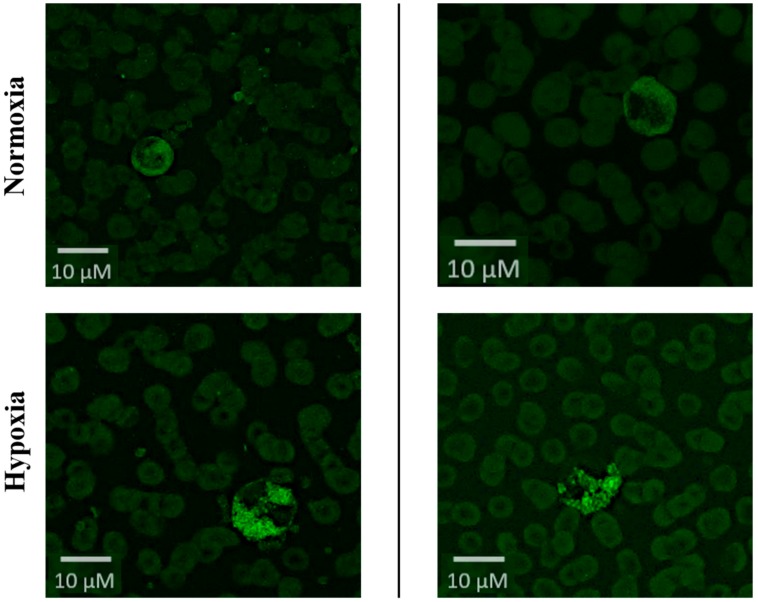
HIF-1α protein visualization. Fluorescence microscopy from peripheral blood smears of 2 randomly chosen individuals at baseline conditions (Normoxia in Munich, 520 m a.s.l, upper part) and hypobaric hypoxia (after 24 h at Little Matterhorn summit, 3.883 m a.s.l., lower part) conditions using a primary mouse antihuman HIF-1α antibody. Erythrocytes can be identified due to their homogenous size (approximately 8 μm), central concavity, and typical non-fluorescent background staining pattern. All other cells with a size of 8–12 μm can be categorized as HIF-1α positive (entirely fluorescent cells), intermediate (partly fluorescent cells), or negative (nonfluorescent cells).

**Figure 2 ijms-21-01034-f002:**
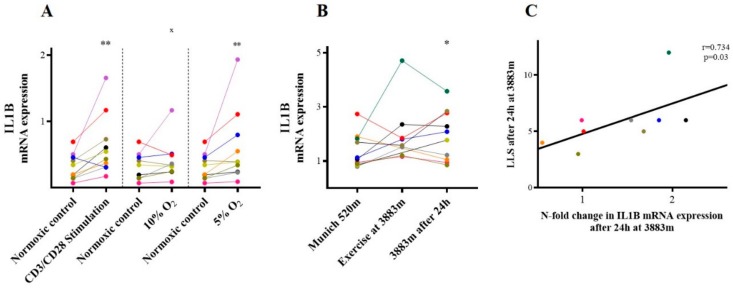
IL-1β. IL-1β mRNA expression (**A**) in vitro following inflammatory and hypoxic incubation, and (**B**) in vivo at different timepoints. (**C**) Association between LLS and IL-1β mRNA expression (n-fold change) at 3883 m a.s.l. after 24 h. * *p* ≤ 0.05 and ** *p* < 0.01 vs. baseline (Munich 520 m or normoxic control in vitro), ^x^
*p* ≤ 0.05 vs. CD3/CD28-stimulation, *p* ≤ 0.05 vs. prior time point in vivo (n.s.). IL-1β = Interleukin 1β. LLS = Lake Louise Score.

**Figure 3 ijms-21-01034-f003:**
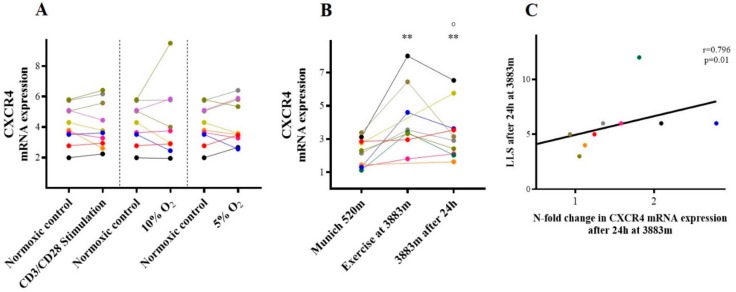
CXCR-4. CXCR-4 mRNA expression (**A**) in vitro following inflammatory and hypoxic incubation, and (**B**) in vivo at different timepoints. (**C**) Association between LLS and CXCR4 mRNA expression (n-fold change) at 3883 m after 24 h. ** *p* < 0.01 vs. baseline (Munich 520 m a.s.l. or normoxic control in vitro), ° *p* ≤ 0.05 vs. prior time point in vivo. CXCR4=C-X-C chemokine receptor type 4.

**Figure 4 ijms-21-01034-f004:**
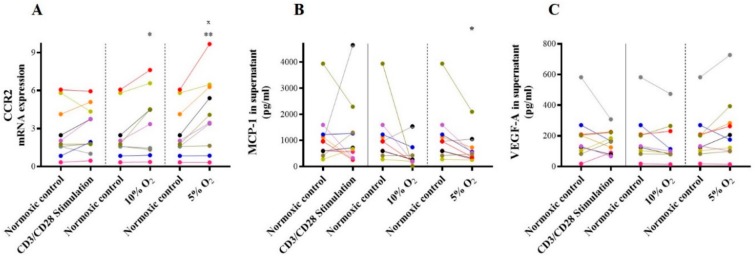
CCR-2, MCP-1, and VEGF-A in vitro. In vitro mRNA expression of (**A**) CCR-2 and (**B**) MCP-1, as well as (**C**) VEGF-A supernatant concentration, both following inflammatory and hypoxic incubation; * *p* ≤ 0.05, ** *p* < 0.01, and ^x^
*p* ≤ 0.05 vs. CD3/CD28-stimulation, CCR-2=C-C chemokine receptor type 2. MCP-1 = Monocyte chemotactic protein 1. VEGF-A = vascular endothelial growth factor A.

**Figure 5 ijms-21-01034-f005:**
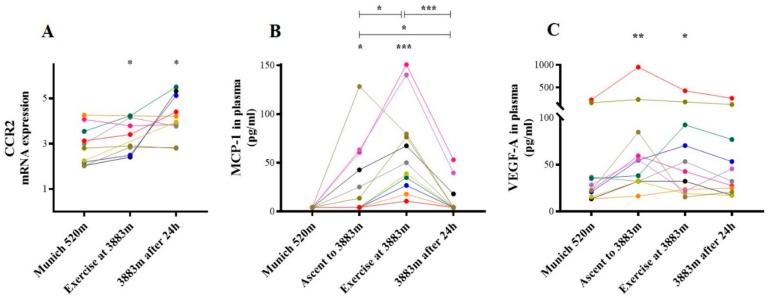
CCR-2, MCP-1, and VEGF-A in vivo. In vivo mRNA expression of (**A**) CCR-2 and (**B**) MCP-1, as well as (**C**) VEGF-A plasma concentration at time points analyzed; * *p* ≤ 0.05, ** *p* < 0.01, and *** *p* < 0.001 vs. baseline (Munich 520 m a.s.l.). CCR-2=C-C chemokine receptor type 2. MCP-1 = Monocyte chemotactic protein 1. VEGF-A = Vascular epithelial growth factor A.

**Figure 6 ijms-21-01034-f006:**
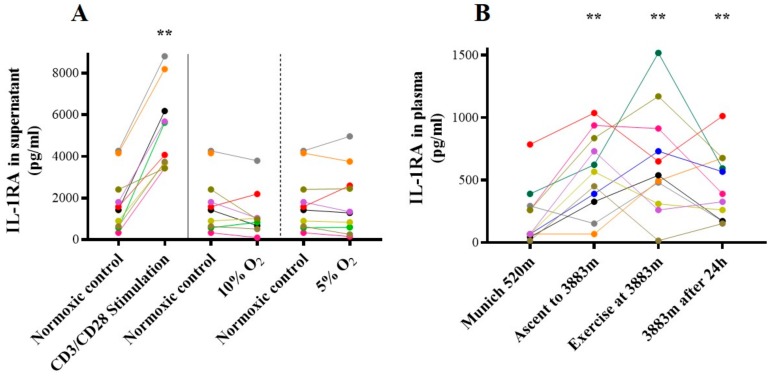
IL-1RA. Concentration of IL-1RA (**A**) in vitro in cell culture supernatant following inflammatory and hypoxic incubation, and (**B**) in vivo in plasma at different timepoints. ** *p* < 0.01 vs. baseline (Munich 520 m a.s.l. or normoxic control in vitro), IL-1RA= Interleukin-1 receptor antagonist.

**Figure 7 ijms-21-01034-f007:**
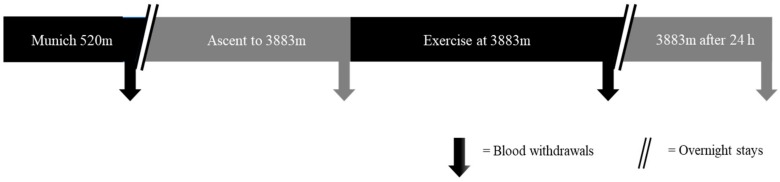
Study protocol for expedition trial. Blood withdrawal and clinical analyses were performed at baseline (Munich 520 m a.s.l.), upon arrival at 3883 m a.s.l. (Ascent to 3883 m), following 90 min of endurance training (Exercise at 3883 m) and after 24 h at 3883 m a.s.l (3883 m after 24 h).

**Table 1 ijms-21-01034-t001:** Volunteer characteristics.

Volunteer Characteristics	*N* = 11
Gender, female/male (numbers; %)	5/6 (45/55)
Age (Years; MV ± SD)	36.4 ± 7
Height (cm; MV ± SD)	178 ± 6
Bodyweight (kg; MV ± SD)	72.7 ± 10
Body Mass Index (kg/m^2^; MV ± SD)	22.7 ± 2
Heart rate (min^−1^; MV ± SD)	61 ± 9
Peripheral oxygen saturation	97 ± 1
(SpO_2;_ %; MV ± SD)	

MV = mean value; SD = standard deviation.

**Table 2 ijms-21-01034-t002:** Primer sequences, UPL Probes, and Real Time Ready Single Assays used in qPCR.

Genes	Primer Sequences, UPL Probes and Assays
TBP forward	5′ GAACATCATGGATCAGAACAACA 3′
TBP reverse	5′ ATAGGGATTCCGGGAGTCAT 3′
	UPL Probe Nr. 87
SDHA forward	5′ GAGGCAGGGTTTAATACAGCA 3′
SDHA reverse	5′ CCAGTTGTCCTCCTCCATGT 3′
	UPL Probe Nr. 132
CCR-2 forward	5′ TGAGACAAGCCACAAGCTGA 3′
CCR-2 reverse	5′ TTCTGATAAACCGAGAACGAGAT 3′
	UPL Probe Nr. 56
IL-1β forward	5′ GAGGCACAAGGCACAACAG 3′
IL-1β reverse	5′ CCATGGCTGCTTCAGACAC 3′
	UPL Probe Nr. 41
CXCR-4	Assay ID 110817
VEGF-A	Assay ID 140396

CXCR-4 = C-X-C Chemokine receptor type 4; CCR-2 = C-C Chemokine receptor type 2; IL1B = Interleukin 1β; SDHA = Succinate dehydrogenase complex, subunit A; TBP = TATA-box binding protein; UPL = Universal Probe Library. VEGF-A=Vascular epithelial growth factor A.

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
