# Peer review of "Hypoxic-Inflammatory Responses under Acute Hypoxia: In Vitro Experiments and Prospective Observational Expedition Trial"

_ijms, 2020, doi:10.3390/ijms21031034_

Round 1

Reviewer 1 Report

A primary contribution of the study by Kammerer et al. is establishment of the tenet that acute hypoxia is a stressor capable of inducing specific inflammatory responses.   In this study, these responses are observed both in vitro and in vivo.  Moreover, they encompass multiple aspects of inflammation, involving both pro- and anti-inflammatory markers and selected components of cytokine/chemokine pathways.  One noteworthy caveat emerging from this investigation is that physiological responses cannot be directly extrapolated from observations conducted in cell culture systems.

As indicated by the authors, this observational study was conducted for the purpose of initially identifying molecular and cellular components of hypoxic responses that can later be evaluated mechanistically. In this multifaceted manuscript, multiple comparisons are evaluated, including in vitro vs in vivo paradigms, immunological vs hypobaric stimuli, pre- vs post-exercise responses, baseline vs high altitude effects and induction of pro- vs anti-inflammatory cytokines.  This complexity necessitates careful establishment of rationale for each comparison.  Additionally, for each comparison there must be parallel presentation of results along with a carefully constructed Discussion in order to adequately interpret study results and appropriately derive of study Conclusions. 

In general, the authors have been successful in organizing and presenting this study. Specifically, procedures are described clearly in Methodology and data are interpreted appropriately in the Results section. However, the Manuscript text needs significant clarification.  While the challenges of conveying information in another language are recognized (and appreciated), it is important that the study’s message be clearly and precisely conveyed. For this reason, recommendations for text revisions are addressed below.

Data presentation:

First, however, consideration of one aspect of data depiction would significantly add to the impact of this study. Specifically, Figures 2 – 7 portray stressor mRNA and cytokine release responses observed for each of the (8 to) 11 participants with statistical significances calculated for each factor using the aggregate subject population.  However, in some cases, there appear to be differing responses for sub-groups, e.g. (at least) 3 subjects show decreases in a) IL-1B mRNA in Figs. 3 A and B; b) CXCR-4 mRNA in Fig. 4A; c)  MCP-1 in Fig. 5B; and d) VEGF-A in Fig. 5C. -while remaining participants show increases (or no change).  Because it is not known whether the same individuals demonstrate similar responses, e.g. increasing levels of multiple stimuli, or whether individuals exhibit varying responses for multiple stimuli, the authors should examine data for each participant to determine whether specific subjects consistently exhibit a certain response, e.g. greater reaction to exercise at high altitude. 

Suggestion: Identifying each individual’s responses using a separate color (that is maintained throughout multiple analyses). This would permit (qualitative) discrimination of whether subtypes of genetic, gender, or fitness backgrounds exist that may be used to stratify physiological responses.  (Note: This was done for subgroups for IL-1RA in Section 4.5.2 and in the companion study Kammerer et al. (2018) for comparisons of subgroup LLS responses to  hypobaric hypoxia where significant differences in rScO2).

Other general editing recommendations:

Extensive yellow highlighting needs to be removed from manuscript

Abbreviations should be defined at first usage, notably for PBMC; a.s.l, MCP-1, HH, LLS

Need to delineate components of comparisons –there are several misuses of ‘respectively’. There are also misuses of ‘e.g.’ (exempli gratia should be used when examples are being provided) and ’i.e.’ (id est should be used when need to specify ‘that is’ or ‘namely.’)

Specific recommendations for editing in manuscript:

Abstract: In the first paragraph, it is difficult to discern whether 2 or 3 conditions are being compared. Suggested editing of these matters in the first paragraph to:

Mammalian responses to hypoxia, specifically to inflammation and to exercise stress are highly related. Both stressors induce the hypoxia-inducible-factor-1α (HIF-1α) pathway, thereby allowing adaptation to hypoxic and inflammatory conditions. At high altitude, alterations of subjects’ HIF, and HIF-target genes e.g. due to genetic variation, allow improved adaptation to hypoxia and are associated with a lowered incidence of acute mountain sickness (AMS). Although we know that the HIF-pathway is induced by hypoxia in conjunction with inflammation or exercise stimuli, however, little is known about the effect of acute hypobaric hypoxia on regulation of HIF-dependent pathways in healthy individuals, for example, mountaineers. We therefore performed in vitro and in vivo experiments with healthy non-acclimatized subjects to test hypotheses that 1) both hypoxic and inflammatory stimuli induce hypoxic-inflammatory signaling pathways in vitro, 2) similar results are seen in vivo in conditions of hypobaric hypoxia, and 3) induction of HIF-dependent genes is associated with AMS.

Introduction: Similar clarification also needs to be incorporated in the first paragraph of the Introduction. Again, a suggestion: 

Mammalian responses to hypoxia ( , ) in combination with either inflammation or exercise stress( , ) are highly related and induce the same HIF 1a signaling pathway ( - ).

Need to better focus the paragraph on cellular tolerance and role of HIF in sepsis. Mention of genetic variants is distracting – need to incorporate more smoothly.  Also, need to provide rationale for inclusion of chemokines and ligands (as done for selected cytokines).

Next to last (summary) paragraph of Intro needs to be clarified. Suggested revisions:

In summary, we know that the HIF-pathway is induced by hypoxia and by inflammation and exercise stimuli. However, little is known about in vivo responses to hypoxia and exercise on regulation of HIF-dependent pathways in healthy individuals, e.g. mountaineers. In this prospective observational trial, we first analyzed in vitro the effects of hypoxic and inflammatory stimuli on HIF-pathway genes and cytokines released in peripheral blood mononuclear cells (PBMCs) from healthy non-acclimatized volunteers. These volunteers then participated in an expedition trial that involved significant changes in oxygen concentrations in which we again analyzed HIF-pathway genes, inflammatory markers, clinical variables, and LLS at intervals following exposure to acute hypobaric hypoxia.

Materials and Methods

Table 1:   Numerical values in right column should be expressed in the same format as indicated in the left column.  For example, the entry for Age (Years: MV + SD) on the left should be represented as 36.4 + 7 in the right column.

In 3.2, Cell culture experiments, it would improve clarity if cell culture conditions were delineated e.g. using letters: a) without stimulation (control); b) with 1 ml of supernatant…; under hypoxia with c) 10% or d) 5% O2 ……

In 3.3 Expedition trial, In line 5,  indicate time lapsed from initial measurement in hours rather than ‘noon’ (don’t know time baseline measurement was taken).  Line 10, indicate as ‘hypoxic exposure’ (rather than exposition).

3.4 Molecular analyses. As there is no quantitation of HIF protein visualization, the second paragraph on quantification is not warranted, i.e. it can be deleted.  Include micron bars on photographs to insure valid visual cell size comparisons.

3.4 Cytokine measurements – How many replicates were conducted for samples in cytokine assays? Also should identify what MCP-1 is.

Results

4.1 Induction of hypoxic-inflammatory pathways. Abbreviation HH is utilized here (as in previous Kammerer et al. (2018) article) but is not defined.  Either define and continue usage of HH in subsequent text or remove HH abbreviation at this location.

4.2 IL-1B. Would add to third sentence:  Similarly, IL-1B mRNA expression increased after 24 h at hypobaric hypoxia on the mountain.

page 12 Move paragraph “In the next step, we analyzed CXCR-4…. Stress and hypoxia’ to become the intro paragraph for Section 4.3.1.  (Also, remove ‘respectively’ from this sentence.)

In next paragraph, it would be helpful to position identification of the figure or table adjacent in the text to the subject being described: CXCR-4 mRNA (Fig. 41) …. SDF-1 protein (the ligand of CXCR-4, Table 3)

Similarly, on page 14, move sentence “We analyzed monocyte chemoattractant receptor CCR-2 mRNA……” so that it is under the 4.3.3 heading. What is meant by ‘MCP-1, enrolled in hypoxic preconditioning, i.e. in stroke and critical for angiogenesis’.  This needs to be clarified.

Fig. 4C – X axis should read “N-fold change in CXCR4 mRNA expression”

Please explain why there are variable numbers of samples analyzed for different factors, e.g. 8 samples in for Fig. 5, 10 in Fig. 6A, 11 in Fig. 6B, etc.?

4.3.2 Need to re-phrase next-to-last sentence- perhaps something like “Notably, plasma concentrations of VEGF-A were 10-fold greater than other subjects at baseline, and further increased when subjected to hypoxia.

Page 17, move sentence “We analyzed plasma concentrations of additional pro- and anti-inflammatory cytokine proteins” under heading 4.4 Proinflammatory cytokines.

4.4.1 Interleukin-3. First sentence may be better expressed as  ‘… IL-3 plasma concentrations remained unchanged…’

4.4.2 Interleukin-6.   Explain basis for “As expected” and “Surprisingly”, e.g. cite supporting literature?

4.5.1 IL-10. Would include “IL-10 supernatant concentrations from in vitro PBMC cultures (Table 3) increased …..  Would also modify next sentence “In vivo, IL-10 plasma concentrations (<2.12 pg/ml) were greatly increased ..., kept increasing after exercise …, then returned to baseline…

4.5.2 IL-1RA. In second paragraph, do you mean ‘the four individuals with lowest LLS showed double the IL-1RA concentrations compared to those with highest (rather than lowest) LLS?

4.6 Volunteer characteristics. This section should be moved to the Materials and Methods under Section 3.1 (which has the same subtitle)

5.0 Discussion. P. 21 – The paragraph on HIF suppression of Tregs does not complement the data presented or add to interpretation of this study, i.e. it should be removed from the Discussion.

Reviewer 2 Report

The study investigating hypoxic inflammatory response in vitro as well as in human is interesting. However, the relation between HIF-1 protein expression and the increase in cytokines and other related proteins and genes is not clear although the authors demonstrated HIF protein expression under hypobaric and hypoxia conditions in Figure 2. Scheme of possible reaction pathway in vitro and in human under hypobaric and hypoxia conditions should be shown. This may greatly help many readers to understand the rather confusing results in this study.

Under hypoxia conditions, effects of nitric oxide is greatly enhanced and vice versa (Am J Respir Crit Care Med. 2000 162:1257-61). The effects of nitric oxide pathway on inflammatory response under hypobaric and hypoxia conditions should be discussed.

Reviewer 3 Report

The authors compare the response to high altitude hypoxia in the blood of healthy subjects to the response of their PBMCs to in vitro hypoxia and inflammation.

The topic is a nice example of combination of in vitro and in vivo study of pathophysiology.

Nevertheless, there are points which are not completely clear and should be improved.

ABSTRACT AND TEMPLATE: the text is too long. IJMS requires abstracts to be shorter than 200 words. Please also adjust the text to the journal template, especially by avoiding highlights in yellow color. Also, when you put the reference numbers at the end of a sentence, please insert them before the full stop. These mistakes make your work look less valuable, like if it has been submitted in a rush and may create misudertandings on the scientific value of the work which seems interesting. Please also add a list of abbreviations at the end. I would also suggest to use more colors in the figures, in order to improve readability and impact.

INTRODUCTION

For the sentence “Thus, it is known that a single hypoxic or inflammatory stimulus in mammalian cells […] lipopolysaccharid tolerance. [4,7,12]. You are underlining two different concept, please split the sentence in two parts and underline which reference says what. What do you mean by minor stimulus? Doesn’t the “lypopolisaccharid tolerance” only refers to inflammation and not to hypoxia too?

MATERIALS AND METHODS

 “3.1 Volunteer characteristics”: Please move most of this paragraph and table 1 at the beginning of the “results” section. You anyway repeated those concepts in the final paragraph of the Results section. In that position, the sentence “we enrolled […] Table 1” sounds like a repetition, completely disconnected from the general context.

“3.2 In vitro experiments” The timings of the in vitro treatments are not totally clear to me. You say cells were kept 24 h at 37°C with 5%CO2. Is this also the duration of the exposure to the four treatments? Whose PBMC did you previously stimulate with Dynabeads? How long did the stimulation last for? Why did you use supernatant from previously stimulated cells instead of directly stimulating? Please provide a reference or a better explanation of this choice (if possible, through a summarizing figure). If, on the other hand, you believe most people should be familiar with the technique, please add such explanation as supplementary material. I would also shorten the explanation about the choice of 10% hypoxia by moving the calculations section (from “international barometric formula” till the end of the higlated part) to the supplementary materials.

“3.3 Expedition trial”: your volunteers have been exercizing by going up and down different altitudes, as they were skiing. Did you consider the fact that they were changing their altitude while exercizing? Did you count how many times they descended and reascended during those 120 minutes? For next studies, it may be interesting to keep the subjects at the same altitude while exercizing, in order to avoid the possible effects of the 300 m altitude gap.

RESULTS:

FIGURES: From the figures, at a first look, your control seems to be a measure of the situation BEFORE the treatment, while, as far as I could understand, it refers to untreated PBMCs, after the same time of incubation as the treated cells. I would suggest, if possible, to rename control as “untreated ” or something like“normoxic” and reorganize the figures in order to avoid the repetition of the “control” column next to every condition.

Table 3: It contains some nice significant results that may look more interesting if represented graphically. If possible, I would leave in form of table just the parameters that does not significantly change. It would give a better visible impact to the article

DISCUSSION:

Please put “limitations” and “conclusions” in separate sections. In particular, I would put together the paragraph “Thus, in this prospective observational study…” with the last paragraph (“In conclusion”) as they seem to express similar concepts.

Kind Regards

Round 2

Reviewer 1 Report

This version of the manuscript is considerably improved as the authors have responded conscientiously and constructively to all three reviewers' comments. Clarity and focus have been significantly improved in the revised manuscript. 

With regard to suggestions to view the data individually - possibly by separate colors or design of data points - to ascertain whether results can be (qualitatively) recognized as aggregating in defined strata or groups, the authors have reply that they have analyzed for statistical outliers- leading one to wonder whether they may not understand the suggestion. However, this is an issue to determine whether there may be additional information to be gleaned from the data points - but does not negate the results presented. 

Author Response

Dear Reviewer!

We would like to thank you again for the new review work. Please find our comments below: 

"With regard to suggestions to view the data individually - possibly by separate colors or design of data points - to ascertain whether results can be (qualitatively) recognized as aggregating in defined strata or groups, the authors have reply that they have analyzed for statistical outliers- leading one to wonder whether they may not understand the suggestion. However, this is an issue to determine whether there may be additional information to be gleaned from the data points - but does not negate the results presented."

Authors comment: Due to your suggestions, we have now re-designed every figure and marked volunteers with an individual color to facilitate identification. As you have correctly noted, this will make the graphics much clearer. We have also added this information in the Statistics section. 

Best regards,

Tobias Kammerer and Simon Schäfer

Reviewer 2 Report

Manuscript has been revised very well. I have no further comments and criticism.

Author Response

Thank you again for your suggestions and the review work.

Best regards,

Tobias Kammerer and Simon Schäfer

Reviewer 3 Report

I still have a few suggestion for the Authors:

The Authors still need to reduce the length of the abstract. 378 words, when the word limit is 200, is still too much.

I also still believe the use of colors in the figure would help through the reading. This may help through the comparison of different figures and to see if the outliners are the same at any condition. Changing colors or symbols in an already plotted graph on GraphPad Prism does not require much time if you still have the original file.

Concerning the methods section, when describing the condition “CD3/CD28" you should say clearly that the previously stimulated PBMCs I suggest to state clearly that those PBMCs were from the same subject.

Conclusion: In the following sentence I would state clearly that the conclusion involves in vitro models involving PBMCs (for example by adding the bold part: "In detail, exposure of healthy volunteers to acute hypobaric hypoxia with endurance exercise had a different and more pronounced effect on mRNA expression and cytokine release from PBMCs than isolated inflammatory or hypoxic incubation in vitro."

Kind regards

Author Response

Dear Reviewer,

thank you again for the suggestions and recommendations! Again, changes are marked in yellow in the word file.

"The Authors still need to reduce the length of the abstract. 378 words, when the word limit is 200, is still too much."

Authors comment: I have reduced the abstract from 378 to 200 words now, hoping that no information has been lost.

"I also still believe the use of colors in the figure would help through the reading. This may help through the comparison of different figures and to see if the outliners are the same at any condition. Changing colors or symbols in an already plotted graph on GraphPad Prism does not require much time if you still have the original file."

Authors comment: We have now re-designed every figure and marked volunteers with an individual color to facilitate identification. We have also added this information in the Statistics section. 

"Concerning the methods section, when describing the condition “CD3/CD28" you should say clearly that the previously stimulated PBMCs I suggest to state clearly that those PBMCs were from the same subject."

Authors comment: we have stated this now clearly in the Methods section.

"Conclusion: In the following sentence I would state clearly that the conclusion involves in vitro models involving PBMCs (for example by adding the bold part: "In detail, exposure of healthy volunteers to acute hypobaric hypoxia with endurance exercise had a different and more pronounced effect on mRNA expression and cytokine release from PBMCs than isolated inflammatory or hypoxic incubation in vitro."

Authors comment: we have revised this sentence in the Conclusion section to state, that we refer to PBMCs. 

Best regards,

Tobias Kammerer and Simon Schäfer

Round 3

Reviewer 1 Report

This revised version represents a concerted effort to convey maximal information from this study and to provide readers with optimal opportunity to review and assess the data.

Reviewer 3 Report

The form and quality have singnificantly improved through these revisions. I accept the manuscript in the present form.

Kind regards

This manuscript is a resubmission of an earlier submission. The following is a list of the peer review reports and author responses from that submission.

Round 1

Reviewer 1 Report

In this study, authors used 11 healthy subjects’ PBMCs treated with hypoxia (5% O2) or CD3/CD28 (as inflammatory) condition to study cytokine mRNA expression by RT-PCR and cytokine releases by ELISA. The authors then tested healthy subjects under hypobaric hypoxia condition (3883m) before and after exercise, as well as 24hrs after hypobaric hypoxia. The changes of gene expression (e.g. IL-6, IL-10, CCR-2, CXCR-4, IL-1b etc.) were examined and reported. The authors concluded that “hypoxia-inflammation pathways are differentially regulated…”. Unfortunately, this study only reported the changes of various gene expression patterns with no related underlying molecular mechanism(s). In addition, there are errors found in the manuscript such as no Fig. 1 and Table 1.

Reviewer 2 Report

In the present study, Kammerer and colleagues have examined how in vitro hypoxia compared to vivo hypobaric hypoxia alters pro-inflammatory responses. For the in vitro experiments, they have isolated PBMCs from 11 healthy volunteers and subjected them to 5% or 10% Ocompared to normoxic conditions. For the in vivo studies, the authors analyzed whole blood RNA and plasma cytokines from the same volunteers upon arrival at 3883 m, 90 min of exercise at 3883m and after 24h at 3883 m. Furthermore, the authors assessed how these findings associate with the clinical presentation of acute mountain sickness. 

Critique.

This is an interesting study, which provides novel findings on the effects of hypoxia in pro-inflammatory responses. I have the following suggestions.

Major points.

1.    The authors emphasize the role of HIF as an upstream regulator of hypoxia-induced alterations in inflammatory responses but data of HIF activation in their experimental system are lacking. The reviewer thinks that it would be helpful to assess HIF1 and HIF2 protein levels to examine whether the degree of hypoxia used was sufficient to stabilize HIF signaling. Furthermore, they could assess the mRNA levels of hypoxia regulated genes (for example PGK1GLUT1)

2.    Could the authors provide information how they selected the oxygen levels used for their in vitro studies? For the altitude of 3883m a.s.l, the corresponding oxygen level is ~13%. Therefore, it would be perhaps more appropriate if they had included this oxygen level for their in vitro experiments.

3.    The reviewer thinks that the authors should take into consideration the difference in the two settings they compare. Specifically, for their in vitro studies they used PBMCs whereas for the in vivo studies they used whole blood and plasma and therefore other cell types may contribute to the observed responses. This limitation should be appropriately discussed.

4.    In addition to the pro-inflammatory effects of hypoxia, there is a significant amount of literature showing that HIFs may induce immunosuppressive responses. Therefore, the authors could provide a more balanced view by including some of these studies in their manuscript.